# Diffusion Barrier Properties of the Intermetallic Compound Layers Formed in the Pt Nanoparticles Alloyed Sn-58Bi Solder Joints Reacted with ENIG and ENEPIG Surface Finishes

**DOI:** 10.3390/ma15238419

**Published:** 2022-11-26

**Authors:** Hyeokgi Choi, Chang-Lae Kim, Yoonchul Sohn

**Affiliations:** 1Department of Welding and Joining Science Engineering, Chosun University, Gwangju 61452, Republic of Korea; 2Department of Mechanical Engineering, Chosun University, Gwangju 61452, Republic of Korea

**Keywords:** intermetallic compound, diffusion barrier, interfacial reaction, Sn-58Bi, scanning electron microscopy

## Abstract

Pt-nanoparticle (NP)-alloyed Sn-58Bi solders were reacted with electroless nickel-immersion gold (ENIG) and electroless nickel-electroless palladium-immersion gold (ENEPIG) surface finishes. We investigated formation of intermetallic compounds (IMCs) and their diffusion barrier properties at reaction interfaces as functions of Pt NP content in the composite solders and duration of solid-state aging at 100 °C. At Sn-58Bi-xPt/ENIG interfaces, typical Ni_3_Sn_4_/Ni_3_P(P-rich layer) microstructure was formed. With the large consumption of the Ni-P layer, the Ni-P and Cu layers were intermixed and Cu atoms spread over the composite solder after 500 h of aging. By contrast, a (Pd,Ni)Sn_4_/thin Ni_3_Sn_4_ microstructure was observed at the Sn-58Bi-xPt/ENEPIG interfaces. The (Pd,Ni)Sn_4_ IMC effectively suppressed the consumption of the Ni-P layer and Ni_3_Sn_4_ growth, functioning as a good diffusion barrier. Therefore, the Sn-58Bi-xPt/ENEPIG joint survived 500 h of aging without microstructural degradation. Based on the experimental results and analysis of this study, Sn-58Bi-0.05Pt/ENEPIG is suggested as the optimum combination for future low-temperature soldering systems.

## 1. Introduction

These days, low-temperature soldering is an important issue in terms of both reduction of CO_2_ emission and mechanical reliability of microelectronic packages. Engineers are researching more environmentally friendly process and trying to lower the bonding temperature within a range that does not impair the mechanical reliability of electronic components. The board to package warpage problem that occurs during the reflow process can be significantly reduced by lowering bonding temperature by using low-temperature solders such as Sn-58Bi, Sn-52In, and Sn-9Zn [1,2]. Eutectic Sn-58Bi alloy with a melting point of 139 °C has been widely studied and used for low-temperature soldering [3]. However, Sn-58Bi solder has some disadvantages, such as low ductility and poor fatigue property. In addition, Bi segregation around intermetallic compound (IMC) layers on Cu metallization was reported to be closely related with embrittlement phenomena of Sn-Bi solder joints [4,5]. To overcome this embrittlement and to enhance mechanical properties, various fillers have been added into Sn-58Bi solder. They included alloying elements such as Cu, Ni, Ag, In, and Sb and nanoparticles such as carbon nanotube, graphene, Y_2_O_3_, Al_2_O_3_, and TiO_2_ [6,7,8,9,10,11,12].

Electroless Ni-electroless Pd-immersion gold (ENEPIG) plating system was developed to replace conventional Electroless Ni-immersion gold (ENIG) used for surface finish layer of printed circuit board (PCB). The ENIG surface treatment has the problem of black pad and brittle fracture [13,14,15]. The black pad refers to a phenomenon in which the nickel layer is corroded by the potential difference during gold plating, which can be prevented by inserting a Pd layer in the middle. The brittle fracture is known to occur through Kirkendall voids in the Ni-Sn-P layer formed during the reaction of electroless Ni-P and Sn-based solders. It has been reported that formation of the Ni-Sn-P layer can also be significantly retarded by a formation of PdSn_4_-type IMC during the reaction between electroless Pd and Sn-based solders. Experiments to verify the usefulness and mechanical reliability of ENEPIG have been performed with various kinds of solders. Most of the studies were performed with Sn-3.0Ag-0.5Cu solder [16,17,18,19,20,21,22,23,24] and a small number of studies were performed with Sn-Ag [25,26] and Sn-Bi alloys [27,28]. Through these experiments, it was proven that the use of ENEPIG, instead of ENIG, reduced the consumption of the Ni-P layer and delayed the formation of the Ni_3_SN_4_/Ni-Sn-P microstructure, thereby preventing brittle fracture and improving the mechanical reliability. Peng et al. [16] reported that (Pd,Ni)Sn_4_ IMC persisted longer for the reaction of ENEPIG/Sn-37Pb compared with the reaction of ENEPIG/Sn-3.0Ag-0.5Cu. However, in both cases, (Pd,Ni)Sn_4_ IMC was detached from the reaction interface within 90 s. The presence of PdSn_4_-type IMC is of great importance because it governs the change of the interfacial microstructure. However, despite this importance, there have been few reports on the behaviors of PdSn_4_ IMC during solid-state aging.

Though numerous filler materials have been tried into lead-free solders, Pt particles modified solder alloy has not been reported yet. A comparative study was conducted to investigate the diffusion barrier properties of interfacial IMCs formed at Pt particles modified Sn-58Bi solders reacted with ENIG and ENEPIG. The sequence of IMC formation and consumption of the Ni-P layer were analyzed as functions of solder composition and duration of solid-state aging. The formation and sustainability of the (Pd,Ni)Sn_4_ compound were also traced during the solid-state aging experiments. An optimal combination of solder composition and type of surface finish is proposed based on the results and analysis.

## 2. Materials and Methods

### 2.1. Materials and Specimen Preparation

To fabricate Sn-58Bi-xPt composite solders, commercial Sn-58Bi solder paste (BBI-LFSP04 from BBIEN, metal particle: 25–45 μm) was mixed with Pt nanoparticles (NPs) (20–30 nm; Avention, Republic of Korea) using a paste mixer (PDM 300V, KM tech Co., Republic of Korea). One minute of planetary mixing was repeated twice for each composition. The planetary mixing conditions were set at a revolution speed of 800 rpm and rotation speed of 500 rpm. The Pt NPs content in the solders was varied as 0, 0.05, 0.1, 0.2 and 0.5 wt%. In this study, the fabricated composite solders were denoted as Sn-58Bi-xPt, where x is the Pt NPs content in wt% base relative to the weight of Sn-58Bi. 

The fabricated Sn-58Bi-xPt solder pastes were printed on the PCB substrates plated with ENIG or ENEPIG surface finishes. The diameter of each metal pad in the flame retardant 4 (FR-4) PCB was 300 μm. The ENIG surface finish consisted of 5 μm-thick Ni-P (5.0–7.0 wt.% P) and 0.05 μm-thick Au. The ENEPIG surface finish was composed consecutively of 8 μm-thick Ni-P, 0.08 μm-thick Pd-P (~3.0 wt.% P), and 0.05 μm-thick Au. After printing of Sn-58Bi-xPt solder pastes, soldering reactions were conducted with each surface finish through a reflow process. The reflow was conducted for 550 s with a peak temperature of 180 °C (1809EXL; Heller, USA). The duration time above the solder melting temperature was 135 s. After the reflow process, the specimens were cooled to room temperature, followed by the removal of the residual flux with deflux solution and ethanol. Some of the fabricated specimens were stored in a convection oven for solid-state aging for 100, 200, and 500 h at 100 °C. Figure 1 presents a schematic of the ENIG and ENEPIG soldering configurations and the reflow profile used for specimen fabrication.

### 2.2. Characterization

Metallographic cross-sections of the reacted specimens were prepared to characterize the microstructure and measure the IMC thickness. Cross-sectional micrographs of the specimens were acquired from the secondary electron signals of a field-emission scanning electron microscope (FE-SEM), JSM-7900F from JEOL, with an acceleration voltage of 15.0 kV. The composition and phases of the IMCs were identified using energy-dispersive X-ray (EDX) spectroscopy. The thickness of the IMC in the scanned micrographs was measured using an image analysis software. The thickness of the layer was defined as the total area occupied by the phase divided by its length. The average values were obtained after measuring five different regions for each reaction specimen. 

## 3. Results and Discussion

### 3.1. Microstructure of Sn-58Bi-xPt/ENIG Interface

Figure 2 shows the cross-sectional SEM images of the Sn-58Bi-xPt solder joints that reacted with the ENIG surface finish. Without the addition of Pt NP, a well-known Ni_3_Sn_4_/Ni_3_P(P-rich layer) microstructure was observed on the unreacted Ni-P layer. After the reflow process, a thin Au layer was rapidly dissolved into the solder, and needle-type Ni_3_Sn_4_ IMCs were formed at the interface. As the solid-state aging time increased, the needle-type morphology of Ni_3_Sn_4_ changed to a chunk-type morphology, and the Ni_3_P layer thickened. The Ni-Sn-P layer was supposed to be present between the Ni_3_Sn_4_ and Ni_3_P layer [15,16,17,18,19,20,21,22]. However, after solid-state aging, it was not distinguished without the spalling of Ni_3_Sn_4_ IMCs. With the addition of Pt NP to the Sn-58Bi solder, the growth of Ni_3_Sn_4_ IMCs was suppressed, whereas the overall microstructure remained unchanged. This is consistent with previous reports on composite solders containing various NP fillers [6,7,8,9,10,11,12]. The growth behavior of the interfacial IMCs is discussed further in the next section. 

Figure 3 displays the EDX area mapping results of the Sn-58Bi-0.1Pt/ENIG interface after 200 h of aging. Here, we confirmed that chunk-type Ni_3_Sn_4_ was formed on a thin Ni_3_P layer, with the remaining Ni(P) layer. However, after 500 h of aging, degradation of the solder joints was observed. Figure 4 shows the microstructure of the Sn-58Bi-0.1Pt/ENIG reaction interface after 500 h of aging, in addition to the EDX area mapping results. Here, the Ni-P and Cu layers were totally intermixed. Furthermore, we showed that Cu atoms passed through the Ni-P layer and spread over the composite solder. Therefore, P atoms were located along the Cu-Ni intermixed layer, while some Cu atoms were detected inside the solder layer as shown in the EDX result. This phenomenon indicates that ENIG does not function properly as a diffusion barrier under the given experimental conditions.

### 3.2. Microstructure of Sn-58Bi-xPt/ENEPIG Interface

Figure 5 shows cross-sectional SEM images of Sn-58Bi-xPt solder joints reacted with ENEPIG surface finish. The interfacial microstructures on the ENEPIG were clearly different from those on the ENIG. In all specimens, branch-like (Pd,Ni)Sn_4_ IMCs were formed on thin layer-type Ni_3_Sn_4_ IMC layers. Here, thin Ni_3_P layers were visible underneath the Ni_3_Sn_4_ IMCs only after 500 h of aging and the Ni-Sn-P layer was not detected in any specimen. Literatures reported that first forming phase was (Pd,Ni)Sn_4_ during the reactions between Sn-based solders and ENEPIG [16,17,18,19,20]. Therefore, the Ni_3_Sn_4_ layer was formed and grew under the branch-like (Pd,Ni)Sn_4_ IMCs. Thicknesses of the Ni_3_Sn_4_ IMCs on ENEPIG were much smaller compared to that formed on ENIG, as shown in Figure 2. With Pt NP addition into the solders, Pt element was detected in most (Pd,Ni)Sn_4_ IMCs, though the content of Pt was quite small, approximately 0.5–0.8 at% Pt in the whole (Pd,Pt,Ni)Sn_4_ compound. Based on these observations, it can be concluded that Ni-P consumption on the ENIG was large owing to the formation of Ni_3_Sn_4_ chunks, whereas that on the ENEPIG was small due to the formation of thin Ni_3_Sn_4_ layers. From the experimental results, it was proven that the (Pd,Ni)Sn_4_ IMC functioned as an effective diffusion barrier to suppress the consumption of Ni-P in the ENEPIG surface finish.

Figure 6 shows the microstructure and EDX line mapping results for the Sn-58Bi-0.1Pt/ENEPIG interface after 200 h of aging. Here, it was confirmed that a branch-like (Pd,Pt,Ni)Sn_4_ IMC was formed on the thin-layer-type Ni_3_Sn_4_ IMC. A microstructure with a similar shape was observed at the Sn-58Bi-0.1Pt/ENEPIG interface after 500 h of aging, as shown in Figure 7. Over the remaining Ni-P layer, a Bi map was found on the opposite side of the Sn map, wheres the Pd map was found on the same side as the Sn map. From the analysis results, it can be inferred that the (Pd,Pt,Ni)Sn_4_ IMCs are located at the overlapping positions of the Pd and Sn maps. Therefore, even after 500 h of aging with ENEPIG, a stable interfacial microstructure was maintained, contrary to the microstructural degradation found in the Sn-58Bi-0.1Pt/ENIG interface after 500 h of aging. 

### 3.3. Diffusion Barrier Properties of ENIG and ENEPIG Surface Finishes

The thicknesses of the Ni_3_Sn_4_ IMCs formed after reflow and solid-state aging were measured and are presented in Figure 8. Without the addition of Pt NP, a large number of Ni_3_Sn_4_ IMCs were formed on the ENIG surface. However, with the addition of 0.05 wt% Pt NPs to the Sn-58Bi solder, the Ni_3_Sn_4_ thickness decreased significantly, dropping to less than half of the case without NP doping. The IMC growth inhibitory effect was the best at 0.05 wt% Pt, but the effect decreased with further increasing the Pt content in the solder. The NPs that do not directly participate in the interfacial reaction has been known to impart their influence on IMC formation as discrete particle, by absorbing preferentially at the interface, hindering the diffusion flux of the substrate and thereby suppressing the IMC growth [29,30]. Same role is expected for Pt NPs in the Sn-58Bi-xPt/ENIG interface. The Ni_3_Sn_4_ thickness on ENIG was significantly increased after 200 h of aging than that after 100 h of aging. Meanwhile, for the Sn-58Bi-xPt/ENEPIG reaction, the Ni_3_Sn_4_ thickness was significantly smaller than that formed in the Sn-58Bi-xPt/ENIG reaction. Suppression of Ni_3_Sn_4_ growth by Pt NP addition was also observed in the ENEPIG. However, in this case, the effect seemed to remain similar despite the increase in Pt content. 

The diffusion barrier properties of the ENIG and ENEPIG surface finishes are closely related to the IMC growth behavior and consumption of the Pd-P and Ni-P layers. Therefore, the correlation between these factors was studied in detail. The Ni-P consumption for the formation of Ni_3_Sn_4_ and (Pd,Ni)Sn_4_ IMCs was analyzed for both Sn-58Bi-xPt/ENIG and Sn-58Bi-xPt/ENEPIG reactions. For the Sn-58Bi-xPt/ENIG interface, neglecting the effect of the Ni_3_P layer, the thickness *Δh* of the Ni-P consumed can be calculated with Equation (1).
(1)$$\Delta h\sim \frac{\frac{4}{3}\pi {\left(\frac{d_{solder}}{2}\right)}^{3}\rho_{solder}C_{s}}{\pi {\left(\frac{d_{pad}}{2}\right)}^{2}{\left(f_{Ni}\cdot \rho \right)}_{Ni\left(P\right)}}+\frac{{\left(f_{Ni}\cdot \rho \cdot V\right)}_{Ni_{3}Sn_{4}}}{{\left(f_{Ni}\cdot \rho \cdot A\right)}_{Ni\left(P\right)}},$$
where $C_{s}$ and *f_Ni_* represent the saturation solubility of Ni in the molten solder and the weight fraction of Ni in Ni-P (0.94) or Ni_3_Sn_4_ (0.27). The symbols *d*, *A*, *V*, and *ρ* represent diameter, area, volume, and density (8.74, 8.25, and 8.65 g/cm^3^ for Sn-58Bi, Ni-P, and Ni_3_Sn_4_ [31]). The first term on the righthand side of Equation (1) represents the amount of Ni dissolved in molten solder, and the second term represents the amount of Ni consumed for Ni_3_Sn_4_ formation. The Ni-P thicknesses consumed after 200 h of aging are listed with the calculation results in Table 1. 

The contribution of Ni to Ni_3_Sn_4_ can be separated from that dissolved in the molten solder. By subtracting the Ni-P consumption for the formaton of Ni_3_Sn_4_ from total consumption of Ni-P measured, the amount of Ni dissolution into the molten solder is obtained. In Figure 9a, black and red lines represent the Ni-P consumption used for Ni_3_Sn_4_ and the molten solder, respectively. Thereafter, $C_{s}$ value can be calculated from the equation by using the amount of Ni-P consumption dissolved into the molten solder. From the calculation, the solubility of Ni in Sn-58Bi-xPt was deduced as 0.13–0.16 wt%. The literature [32] reported that the solubility of Ni in Sn-58Bi was estimated to be about 3.4 wt% at 903 K and 1.0 wt% at 733 K. However, the solubility has not been reported at low-temperature range. The solubility deduced in this study (373 K) seems to be in the plausible range when compared with the values reported in the literature. 

The thickness Δ*h* of Ni-P consumed for the reactions between Sn-58Bi-xPt and ENEPIG can be calculated with Equation (2).
(2)$$\Delta h\sim \frac{\frac{4}{3}\pi {\left(\frac{d_{solder}}{2}\right)}^{3}\rho_{solder}C_{s}}{\pi {\left(\frac{d_{pad}}{2}\right)}^{2}{\left(f_{Ni}\cdot \rho \right)}_{Ni-P}}+\frac{{\left(f_{Ni}\cdot \rho \cdot A\right)}_{\left(Pd,Ni\right)Sn_{4}}}{{\left(f_{Ni}\cdot \rho \cdot A\right)}_{Ni-P}}\cdot \frac{{\left(f_{Pd}\cdot \rho \cdot V\right)}_{Pd-P}}{{\left(f_{Pd}\cdot \rho \cdot A\right)}_{\left(Pd,Ni\right)Sn_{4}}}+\frac{{\left(f_{Ni}\cdot \rho \cdot V\right)}_{Ni_{3}Sn_{4}}}{{\left(f_{Ni}\cdot \rho \cdot A\right)}_{Ni-P}}.$$

Here, the first term on the righthand side indicates the Ni-P thickness required for Ni dissolution into the solder. The second and third terms represent the Ni-P consumption for the formation of (Pd,Ni)Sn_4_ and Ni_3_Sn_4_, respectively. The second term was calculated assuming that all Pd atoms in the Pd-P layer were used to form (Pd,Ni)Sn_4_ compound because Pd has no solubility in Sn or Bi under 200 °C [33,34]. Ho et al. estimated the f_Ni_ value for (Pd,Ni)Sn_4_ as 0.08 [35]. Tian et al. [36] reported ρ of (Pd_0.26_Ni_0.74_)Sn_4_ as 7.64 g/cm^3^. For EDX analysis, Pt was usually detected in the (Pd,Ni)Sn_4_ phase in the form of (Pd,Pt,Ni)Sn_4_. However, it was disregarded for this calculation because the Pt content was small, only 0.5–0.8 at% in the entire (Pd,Pt,Ni)Sn_4_ compound. 

The Ni-P consumption for (Pd,Ni)Sn_4_ and Ni_3_Sn_4_ formation and dissolution into the molten solder was calculated for the Sn-58Bi-xPt/ENEPIG reaction (200 h of aging), and the results are presented in Table 1 and Figure 9b. For the Sn-58Bi-xPt/ENEPIG, the Ni(P) consumption for the formation of Ni_3_Sn_4_ was significantly smaller than that of Sn-58Bi-xPt/ENIG. Furthermore, dissolution of Ni into the molten solder was also smaller for Sn-58Bi-xPt/ENEPIG than for Sn-58Bi-xPt/ENIG reaction, which may be attributed to the sluggish Ni diffusion through effective (Pd,Ni)Sn_4_ diffusion barrier. Here, the suppression of Ni_3_Sn_4_ growth with the addition of Pt NP is also evident in the figure.

A comparative schematic model of the interfacial microstructures after reflow and after aging is provided in Figure 10. The analysis in this study shows that the consumption of the Ni-P layer was significantly larger for Sn-58Bi-xPt/ENIG than for Sn-58Bi-xPt/ENEPIG reaction, proving that the (Pd,Ni)Sn_4_ IMC functioned as an effective diffusion barrier to suppress the Ni_3_Sn_4_ growth. The low partitioning of Ni into (Pd,Ni)Sn_4_ also contributed to the low consumption of the Ni-P layer for the Sn-58Bi-xPt/ENEPIG: f_Ni_ = 0.27 for Ni_3_Sn_4_ whereas 0.08 for (Pd,Ni)Sn_4_. Therefore, from the experimental results, it was proven that ENEPIG was a better diffusion barrier than ENIG for the reaction with Sn-58Bi solder. It was reported that the (Pd,Ni)Sn_4_ IMC detached from the reaction interface within 90 s in both the molten Sn-37Pb and Sn-3.0Ag-0.5Cu solders [14]. However, for solid-state aging in this study, the (Pd,Ni)Sn_4_ IMC survived on the reaction interface, even after 500 h of aging at 100 °C. Increase of aging temperature can affect the growth of interfacial IMCs and the mechanical strength of the composite solders. It was reported that both modulus and strength of the soldering structures decreased with increasing aging temperature, which was attributed to the decrease of the volume fraction of dispersed IMC particles and the dislocation density [37].

## 4. Conclusions

This study investigated the diffusion barrier properties of interfacial IMC layers in the Sn-58Bi-xPt solder joints that reacted with ENIG and ENEPIG surface finishes. The specimens were aged for 100–500 h at 100 °C after reflow process and detailed analyses for their microstructures and diffusion barrier properties were conducted after the reactions. The conclusions drawn are summarized as follows.
(1)A typical Ni_3_Sn_4_/Ni_3_P microstructure was obtained after Sn-58Bi-xPt/ENIG reactions. In this structure, the amount of Ni_3_Sn_4_ formed and the consumption of the Ni-P layer were large. After 500 h of aging, degradation of the interfacial microstructure was observed with intermixing between the Ni-P and Cu layers, where Cu atoms diffused through the Ni-P layer and spread over the composite solders.(2)Branch-like (Pd,Ni)Sn_4_ IMCs were formed on the thin Ni_3_Sn_4_ layer for Sn-58Bi-xPt/ENEPIG reactions. Here, the consumption of the Ni-P layer was significantly smaller compared with that for the Sn-58BixPt/ENIG reaction. The interfacial microstructure of (Pd,Ni)Sn_4_/Ni_3_Sn_4_ survived solid-state aging for 500 h, where diffusion of Cu atoms into the solder was not found.(3)Alloying Pd NPs into Sn-58Bi solders successfully suppressed the overgrowth of interfacial IMCs on both the ENIG and ENEPIG surfaces. However, 0.05 wt% Pd NPs was sufficient and further addition was ineffective for suppressing IMC growth. Based on the experimental results, Sn-58Bi-0.05Pt/ENEPIG is suggested as a optimum solder joint for low-temperature soldering systems.(4)Under solid-state aging conditions, (Pd,Ni)Sn_4_ IMC persisted for over 500 h at the reaction interface as a good diffusion barrier to suppress Ni_3_Sn_4_ growth. The low partitioning of Ni into (Pd,Ni)Sn_4_ contributed to the low consumption of the Ni-P layer for the Sn-58Bi-xPt/ENEPIG. Partitioning of Ni was analyzed to be 27 wt% for Ni_3_Sn_4_ whereas 8 wt% for (Pd,Ni)Sn_4_.

## Figures and Tables

**Figure 1 materials-15-08419-f001:**
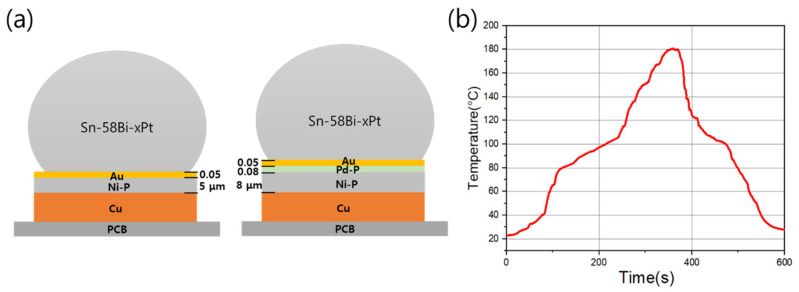
(**a**) Schematic diagram for ENIG and ENEPIG soldering configurations (not to scale); (**b**) reflow profile used for Sn-58Bi-xPt soldering reactions.

**Figure 2 materials-15-08419-f002:**
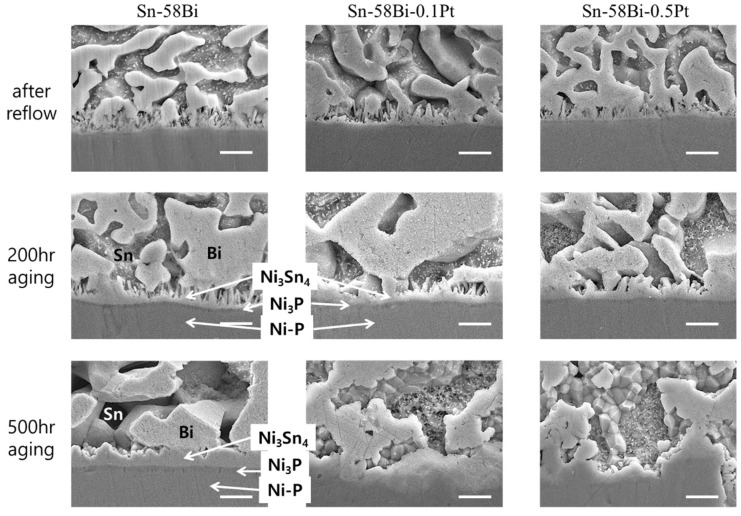
Interfacial microstructures of the Sn-58Bi-xPt/ENIG solder joints. (scale bars: 2 μm).

**Figure 3 materials-15-08419-f003:**
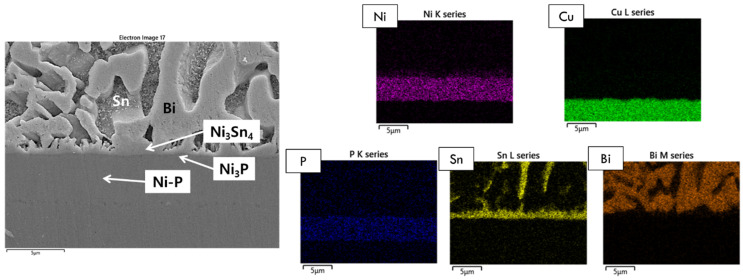
EDX elemental analysis of the Sn-58Bi-0.1Pt/ENIG interface after 200 h of aging.

**Figure 4 materials-15-08419-f004:**
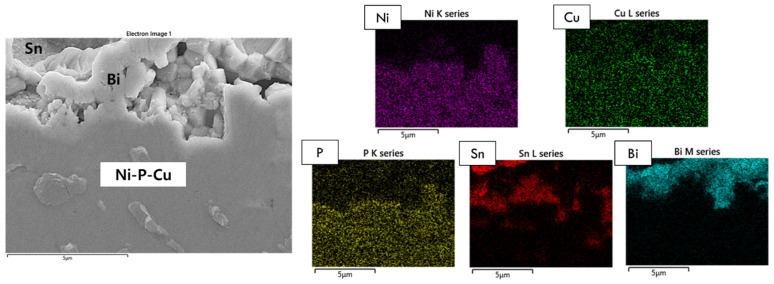
EDX elemental analysis of the Sn-58Bi-0.1Pt/ENIG interface after 500 h of aging.

**Figure 5 materials-15-08419-f005:**
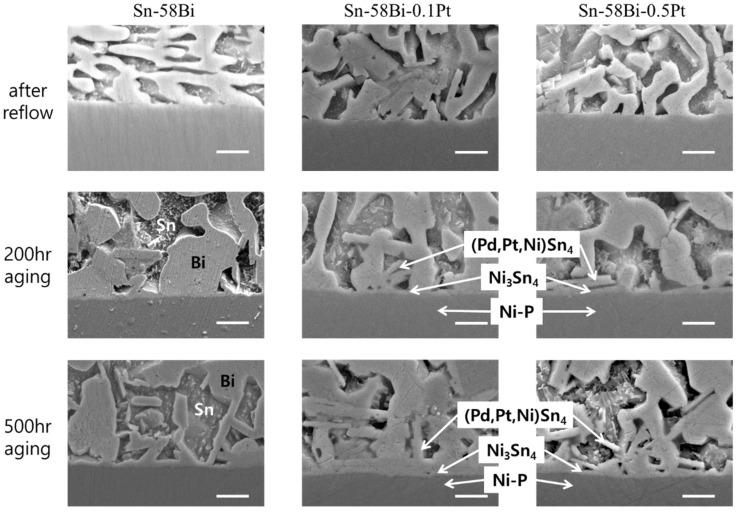
Interfacial microstructures of the Sn-58Bi-xPt/ENEPIG solder joints. (scale bars: 2 μm).

**Figure 6 materials-15-08419-f006:**
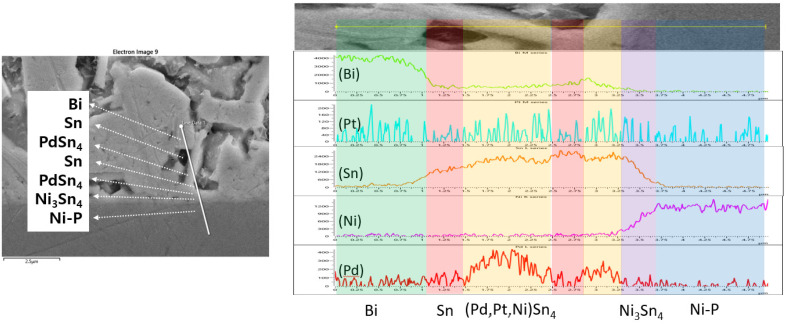
EDX elemental analysis of the Sn-58Bi-0.1Pt/ENEPIG interface after 200 h of aging.

**Figure 7 materials-15-08419-f007:**
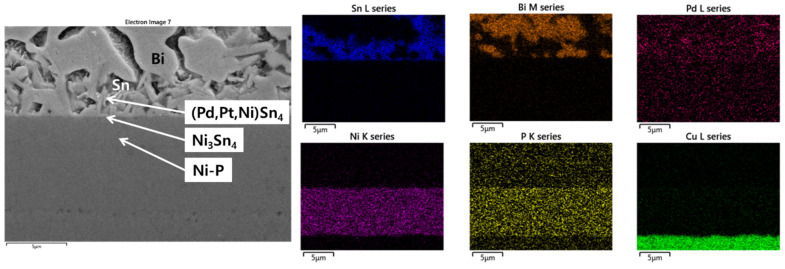
EDX elemental analysis of the Sn-58Bi-0.1Pt/ENEPIG interface after 500 h of aging.

**Figure 8 materials-15-08419-f008:**
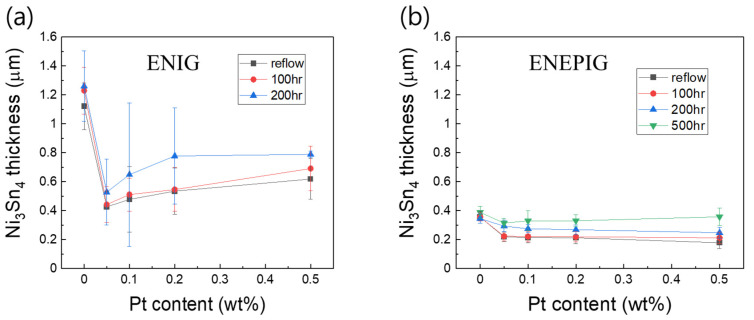
Thickness of Ni_3_Sn_4_ IMCs formed at (**a**) Sn-58Bi-xPt/ENIG and (**b**) Sn-58Bi-xPt/ENEPIG interfaces.

**Figure 9 materials-15-08419-f009:**
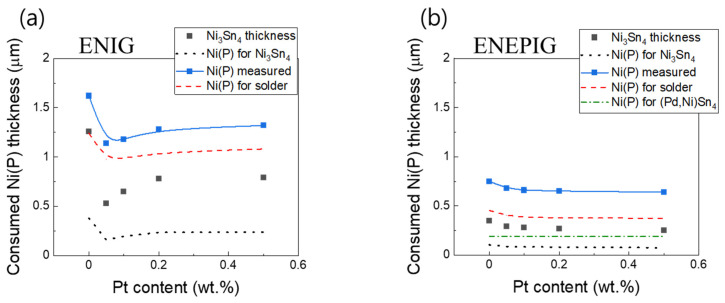
The Ni-P thickness consumed for the formation of interfacial IMCs and dissolution into the molten solder: (**a**) ENIG and (**b**) ENEPIG (200 h of aging).

**Figure 10 materials-15-08419-f010:**
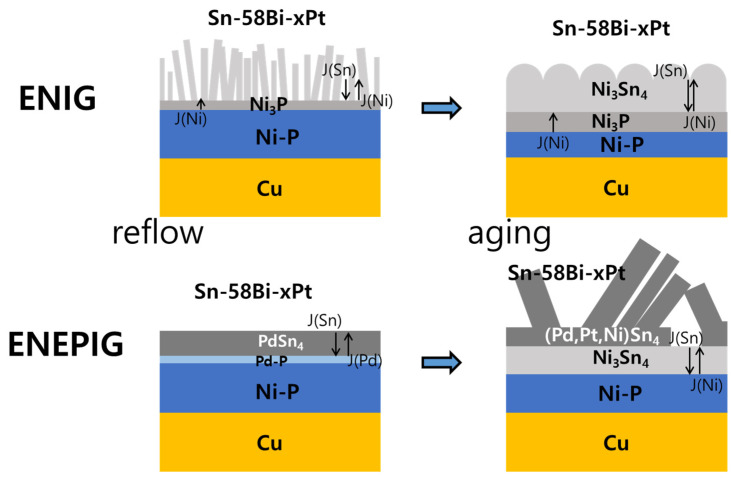
Comparative schematic model of the interfacial microstructures after reflow and after aging with the possible fluxes due to interdiffusion in the ENIG/ and ENEPIG/Sn-58Bi-xPt solder joints. The different components are not to scale.

**Table 1 materials-15-08419-t001:** Consumed thickness of the Ni-P layer for the formation of interfacial IMCs on the ENIG and ENEPIG (200 h of aging).

Surface Finish	IMC Thickness (μm)	Pt NP Content (wt%)				
		0	0.05	0.1	0.2	0.5
ENIG	Ni_3_Sn_4_ (measured)	1.26	0.53	0.65	0.78	0.79
	consumed Ni(P) for Ni_3_Sn_4_	0.38	0.16	0.20	0.23	0.24
	consumed Ni(P) (measured)	1.62	1.14	1.18	1.28	1.32
	Ni(P) dissolution into the solder	1.24	0.98	0.98	1.05	1.08
ENEPIG	Ni_3_Sn_4_ (measured)	0.35	0.29	0.28	0.27	0.25
	consumed Ni(P) for Ni_3_Sn_4_	0.11	0.09	0.08	0.08	0.08
	consumed Ni(P) for (Pd,Ni)Sn_4_	0.19	0.19	0.19	0.19	0.19
	consumed Ni(P) (measured)	0.75	0.68	0.66	0.65	0.64
	Ni(P) dissolution into the solder	0.45	0.40	0.39	0.38	0.37

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
