# Peer review of "Diffusion Barrier Properties of the Intermetallic Compound Layers Formed in the Pt Nanoparticles Alloyed Sn-58Bi Solder Joints Reacted with ENIG and ENEPIG Surface Finishes"

_materials, 2022, doi:10.3390/ma15238419_

Round 1
Reviewer 1 Report
By performing the 500 h of aging, this manuscript concluded that Sn-58Bi-0.05Pt/ENEPIG is the optimum combination for future low-temperature soldering systems, as (Pd,Ni)Sn4 IMC effectively suppressed the consumption of the Ni-P layer and Ni3Sn4 growth. To improve the quality of the manuscript, some comments are suggested as follows:
1. The language can be improved by correct the sentence structure. In addition, more accurate descriptions and discussions should be made especially for the abstract and conclusion.
2. How to make sure the uniformly distribution of Pt nanoparticles in the Sn-58Bi solder paste?
3. In Figure 1, Sn-58Bi-xRu is found rather than Sn-58Bi-xPt.
4. A number of words or symbols are not times new roman.
5. Degradation of the solder joints should be further clarified as it misleads the readers.
6. It is known that the growth of IMC is controlled by the diffusion of atoms. In fact, the aging temperature can affect both liquid and solid states of solder material in terms of tensile deformation and microstructure. The authors are suggested to have some discussions at this aspect. One refence is suggested as: Tensile deformation and microstructures of Sn–3.0Ag–0.5Cu solder joints: Effect of annealing temperature.
7. The Ni-P thicknesses consumed after 200 h of aging are listed with the calculation results in Table 1. What is the reason to only provide the results for 200h?
8. By proposing Eq. (1), the thickness Δh of the Ni-P consumed can be calculated. The solubility of Ni in Sn-58Bi-xPt was deduced as 0.13–0.16 wt% at 373 K. As the temperature is quite different from the literature, it is hard to conclude that the solubility deduced in this study (373 K) was in reasonable agreement with the values reported in the literature. The authors are suggested to make some discussions about the effect of temperature to clarify the reliability of the conclusion.
Author Response
- The language can be improved by correct the sentence structure. In addition, more accurate descriptions and discussions should be made especially for the abstract and conclusion.
English of this paper has been edited by a native speaker. More accurate descriptions were made for the abstract and conclusion following the reviewer’s comment. The authors hope that extra-correction of English may be helped by English editor of Materials.
- How to make sure the uniformly distribution of Pt nanoparticles in the Sn-58Bi solder paste?
Individual Pt nanoparticles could not be detected with our SEM analysis. However, after preparing a paste of specific composition, several parts thereof were collected and DSC experiments of them were performed and compared. It was confirmed that the DSC results were identical.
- In Figure 1, Sn-58Bi-xRu is found rather than Sn-58Bi-xPt.
Sn-58Bi-xPt is correct, which was modified in the figure.
- A number of words or symbols are not times new roman.
The authors just used manuscript template supplied by Materials. We also believe that the words and symbols will further be edited by the editors of Materials.
- Degradation of the solder joints should be further clarified as it misleads the readers.
Degradation of the solder joints was described further as below.
Here, the Ni-P and Cu layers were totally intermixed. Furthermore, we showed that Cu atoms passed through the Ni-P layer and spread over the composite solder. Therefore, P atoms were located along the Cu-Ni intermixed layer, while some Cu atoms were detected inside the solder layer as shown in the EDX result.
- It is known that the growth of IMC is controlled by the diffusion of atoms. In fact, the aging temperature can affect both liquid and solid states of solder material in terms of tensile deformation and microstructure. The authors are suggested to have some discussions at this aspect. One refence is suggested as: Tensile deformation and microstructures of Sn–3.0Ag–0.5Cu solder joints: Effect of annealing temperature.
The reference suggested was introduced into the paper as Ref. 37 with some discussion (red letters in the paper).
- The Ni-P thicknesses consumed after 200 h of aging are listed with the calculation results in Table 1. What is the reason to only provide the results for 200h?
As shown in Fig. 4, degradation of the Sn-58Bi-0.1Pt/ENIG interface occurred after 500 h aging, where measurement of accurate IMC thickness was not possible. Therefore, direct comparison between ENIG and ENEPIG joint was made with 200 h specimen. And the result was summarized in Table. 1.
- By proposing Eq. (1), the thickness Δh of the Ni-P consumed can be calculated. The solubility of Ni in Sn-58Bi-xPt was deduced as 0.13–0.16 wt% at 373 K. As the temperature is quite different from the literature, it is hard to conclude that the solubility deduced in this study (373 K) was in reasonable agreement with the values reported in the literature. The authors are suggested to make some discussions about the effect of temperature to clarify the reliability of the conclusion.
The part was modified with a mild expression as below.
The solubility deduced in this study (373 K) seems to be in the plausible range when compared with the values reported in the literature.

Reviewer 2 Report
In this manuscript, the interfacial behaviors of Pt NPs enhanced Sn-58Bi solder joints with ENIG and ENEPIG surface finishes were investigated. The analysis about diffusion barrier properties is interesting and reasonable. However, there are some comments need to be considered.
1. Please change the title because “Sn-58Bi-xPt” is out of standardization and confusing.
2. Why Pt NPs were chosen as the enhancement phase? The influence mechanism of Pt NPs on the growth of interfacial IMCs should be analyzed.
3. There is a mistake in Figure 1a (Sn-58Bi-xRu?).
4. Does the Pt NPs doping have any other effect on the properties of the solder pastes? Such as wettablity and mechanical properties.
5. Through experimental results and analysis, Sn-58Bi-0.05Pt/ENEPIG is regarded as the optimum combination. However, it needs to be further approved by mechanical property test.
Author Response
In this manuscript, the interfacial behaviors of Pt NPs enhanced Sn-58Bi solder joints with ENIG and ENEPIG surface finishes were investigated. The analysis about diffusion barrier properties is interesting and reasonable. However, there are some comments need to be considered.
- Please change the title because “Sn-58Bi-xPt” is out of standardization and confusing.
The title was corrected as below.
Diffusion barrier properties of the intermetallic compound layers formed in the Pt nanoparticles alloyed Sn-58Bi solder joints reacted with ENIG and ENEPIG surface finishes
- Why Pt NPs were chosen as the enhancement phase? The influence mechanism of Pt NPs on the growth of interfacial IMCs should be analyzed.
The authors expected that Pt atoms may incorporate into PdSn4 IMC as Ni atoms substituted into some of Pd atoms as a form of (Pd,Ni)Sn4, as reported in Ref. 33. As expected, Pd atoms incorporated into the IMC as a form of (Pd,Pt,Ni)Sn4. However, the Pt content was very small, only 0.5-0.8 at% in the entire (Pd,Pt,Ni)Sn4 compound.
For the influence of Pt NPs on IMC growth, the following explanation was added to the paper with additional references [Ref. 29,30].
The NPs that do not directly participate in the interfacial reaction has been known to impart their influence on IMC formation as discrete particle, by absorbing preferentially at the interface, hindering the diffusion flux of the substrate and thereby suppressing the IMC growth [29,30]. Same role is expected for Pt NPs in the Sn-58Bi-xPt/ENIG interface.
- There is a mistake in Figure 1a (Sn-58Bi-xRu?).
Sn-58Bi-xPt is correct, which was modified in the figure.
- Does the Pt NPs doping have any other effect on the properties of the solder pastes? Such as wettablity and mechanical properties.
There was an enhancement of mechanical strength as shown in the figure below. However, the effect was not distinguished and this result was not inserted in this paper.
- Through experimental results and analysis, Sn-58Bi-0.05Pt/ENEPIG is regarded as the optimum combination. However, it needs to be further approved by mechanical property test.
The proposed solder joint is the best combination in terms of interfacial reaction. As in the previous answer, hardness measurement was performed to confirm that there was an increase in hardness, but the effect was not large. Though additional mechanical tests may be conducted later, the results would be published as another paper.

Reviewer 3 Report
In this paper, the authors studied the formation of a diffusion barrier at the interface of Sn-58Bi-xPt solders with ENIG/ENEPIG surface finishes. The paper is interesting and publishable subject to revision.
1.The Introduction is not sufficiently referenced. The authors speak about low temperature solders including Sn-52In and Sn-9Zn, however, references are not provided. The authors should cite papers related to the microstructure and properties of Sn-In and Sn-Zn alloys and their use in soldering, eg, https://dx.doi.org/10.1007/BF03222376, https://doi.org/10.3390/ma15207210.
2.In Fig. 1a, the following chemical composition of the solder is marked: Sn-58Bi-xRu. Is ruthenium a misprint or you had it in the solder?
3.It would be useful to indicate the thickness of the layers in Fig. 1a since the layers were not identical. The thickness of Ni(P) was 5 µm in ENIG and 8 µm in ENEPIG (see the bottom paragraph on page 2). Please, indicate it in the figure itself, i.e., make the Ni(P) layer in ENEPIG larger.
4.The microstructure of the solder alloy before soldering is not shown. Only microstructures after the reflow soldering and ageing are given in Figs. 2-7. It would be useful to see the initial microstructure of the solder.
5.The authors initially judged the diffusion barrier properties of the IMC layer by the amount of Cu diffused from the substrate to the solder (see the end of the second paragraph on page 4 or the conclusion (1)). This assumption is reasonable and justified. However, later only the thickness of the IMC layer was presented and discussed (Fig. 8, Table 1). The concentration of Cu diffused was not compared. The chemical composition of the layers on the ENIG and ENEPIG was not identical. Therefore, it is very difficult to judge the diffusion barrier properties from the thicknesses of the IMC layers alone. It would be more useful to present the concentration of Cu atoms diffused to the solder (concentration of Cu measured by EDS) and compare the diffusion barrier properties of the materials in this way.
6. The different chemical composition of the IMC layer formed at ENIG (Ni3Sn4) and ENEPIG surface finishes ((Pd,Ni)Sn4) should be highlighted. A schematic of diffusion fluxes of the elements across the interface (Sn, Ni, Pd, Pt) and subsequent formation of the IMC layers would be useful and should be provided in the manuscript.
7.The authors should number the lines of the manuscript to facilitate orientation.
Author Response
In this paper, the authors studied the formation of a diffusion barrier at the interface of Sn-58Bi-xPt solders with ENIG/ENEPIG surface finishes. The paper is interesting and publishable subject to revision.
1.The Introduction is not sufficiently referenced. The authors speak about low temperature solders including Sn-52In and Sn-9Zn, however, references are not provided. The authors should cite papers related to the microstructure and properties of Sn-In and Sn-Zn alloys and their use in soldering, eg, https://dx.doi.org/10.1007/BF03222376, https://doi.org/10.3390/ma15207210.
The suggested references were inserted in the manuscript. [Ref. 1, 2]
2.In Fig. 1a, the following chemical composition of the solder is marked: Sn-58Bi-xRu. Is ruthenium a misprint or you had it in the solder?
Sn-58Bi-xPt is correct, which was modified in the figure. (no Ru in the solder)
3.It would be useful to indicate the thickness of the layers in Fig. 1a since the layers were not identical. The thickness of Ni(P) was 5 µm in ENIG and 8 µm in ENEPIG (see the bottom paragraph on page 2). Please, indicate it in the figure itself, i.e., make the Ni(P) layer in ENEPIG larger.
Figure 1 was modified following reviewer’s comment.
4.The microstructure of the solder alloy before soldering is not shown. Only microstructures after the reflow soldering and ageing are given in Figs. 2-7. It would be useful to see the initial microstructure of the solder.
As described in “Materials and method” section, commercial Sn-58Bi solder paste was used to fabricate the composite solders. Size of the Sn-58Bi metal particles was 25-45 μm. Eutectic microstructure of Sn-58Bi can be found in numerous papers.
For the fabrication of the composite solder paste, please refer to the description in Materials and method section.
To fabricate Sn-58Bi-xPt composite solders, commercial Sn-58Bi solder paste (BBI-LFSP04 from BBIEN, metal particle:25–45 μm) was mixed with Pt nanoparticles (NPs) (20–30 nm; Avention, KOREA) using a paste mixer (PDM 300V, KM tech Co., Korea).
5.The authors initially judged the diffusion barrier properties of the IMC layer by the amount of Cu diffused from the substrate to the solder (see the end of the second paragraph on page 4 or the conclusion (1)). This assumption is reasonable and justified. However, later only the thickness of the IMC layer was presented and discussed (Fig. 8, Table 1). The concentration of Cu diffused was not compared. The chemical composition of the layers on the ENIG and ENEPIG was not identical. Therefore, it is very difficult to judge the diffusion barrier properties from the thicknesses of the IMC layers alone. It would be more useful to present the concentration of Cu atoms diffused to the solder (concentration of Cu measured by EDS) and compare the diffusion barrier properties of the materials in this way.
As shown in Fig. 4, degradation of the Sn-58Bi-0.1Pt/ENIG interface occurred after 500 h aging, where measurement of accurate IMC thickness was not possible. Therefore, direct comparison between ENIG and ENEPIG joint was made with 200 h specimens. And the result was summarized in Table. 1. The authors believe that comparison with 200 h specimens is more meaningful and important for the comparison of the two surface finishes because Cu penetration into the solder after 500 h aging means that the Ni-P layer completely lost its function as a diffusion barrier, as obviously shown in Fig. 4. We remark that no Cu was detected inside the solders for both ENIG/ and ENEPIG/Sn-58Bi-XPt specimens until 200 h of aging.
- The different chemical composition of the IMC layer formed at ENIG (Ni3Sn4) and ENEPIG surface finishes ((Pd,Ni)Sn4) should be highlighted. A schematic of diffusion fluxes of the elements across the interface (Sn, Ni, Pd, Pt) and subsequent formation of the IMC layers would be useful and should be provided in the manuscript.
A schematic figure is drawn and provided as Fig. 10 in the paper following reviewer’s comment.
7.The authors should number the lines of the manuscript to facilitate orientation.
Line numbering of the manuscript was displayed following reviewer’s comment.

Round 2
Reviewer 1 Report
The manuscript has been carefully revised by the authors and can be accepted for publication.
Reviewer 2 Report
The authors have made comprehensive revisions based on the comments of the reviewers, and the paper can be considered for acceptance in its current version.
Reviewer 3 Report
Authors answered most of my comments. The paper can be accepted for publication.